# Quorum Sensing Inhibitors from Marine Microorganisms and Their Synthetic Derivatives

**DOI:** 10.3390/md17020080

**Published:** 2019-01-28

**Authors:** Jianwei Chen, Bixia Wang, Yaojia Lu, Yuqi Guo, Jiadong Sun, Bin Wei, Huawei Zhang, Hong Wang

**Affiliations:** 1College of Pharmaceutical Science & Collaborative Innovation Center of Yangtze River Delta Region Green Pharmaceuticals, Zhejiang University of Technology, Hangzhou 310014, China; cjw983617@zjut.edu.cn (J.C.); 15757116051@163.com (B.W.); lyj512637@zjut.edu.cn (Y.L.); m17816038135_1@163.com (Y.G.); binwei@zjut.edu.cn (B.W.); hwzhang@zjut.edu.cn (H.Z.); 2Laboratory of Bioorganic Chemistry, National Institute of Diabetes and Digestive and Kidney Diseases (NIDDK), National Institutes of Health, Bethesda, MD 20878, USA; jiadong.sun@nih.gov

**Keywords:** quorum sensing, quorum sensing inhibitor, marine microorganism, derivatives

## Abstract

Quorum sensing inhibitors (QSIs) present a promising alternative or potent adjuvants of conventional antibiotics for the treatment of antibiotic-resistant bacterial strains, since they could disrupt bacterial pathogenicity without imposing selective pressure involved in antibacterial treatments. This review covers a series of molecules showing quorum sensing (QS) inhibitory activity that are isolated from marine microorganisms, including bacteria, actinomycetes and fungi, and chemically synthesized based on QSIs derived from marine microorganisms. This is the first comprehensive overview of QSIs derived from marine microorganisms and their synthetic analogues with QS inhibitory activity.

## 1. Introduction

We have found ourselves facing a significant problem in modern healthcare settings where many anti-infective drugs have lost their effectiveness against life-threatening and debilitating diseases [1,2]. The pathogens have outplaced our abilities to sustainably manage them. Thus, there is an urgent need to discover new types of antimicrobial compounds and novel mechanisms for disease prevention and treatment. One competitive antimicrobial advantage proposed in targeting quorum sensing (QS) is that the treatment of quorum sensing inhibitors (QSIs) does not inhibit bacterial growth and does not also exert a selective pressure to develop bacterial resistance to this treatment. Granted, interference of QS will likely decrease bacterial fitness for survival under certain conditions, but if a delicate control is performed on pathogenic QS-regulated genes, then developing resistance mechanisms against QS-inhibiting therapies may be a difficult proposition in pathogenic bacteria.

QS is a cell–cell communication process that enables bacteria to regulate their collective behaviors in response to population density and species composition changes in the surrounding environment. It allows bacteria to synchronize gene expression by virtue of extracellular signaling molecules called autoinducers. These autoinducers are released into the surrounding environment where they could be recognized by specific receptors that reside either in the cytoplasm or in the membrane. When autoinducers reach a certain threshold concentration, a signal cascade is triggered that promotes synchronous gene expression in the population of bacteria, such as bioluminescence, the secretion of virulence factors, the formation of biofilm, and other biological behaviors. Commonly, gram-negative bacteria use the cytoplasmic transcription factors, LuxR-type QS receptors, to detect *N*-acyl-homoserine lactones (AHLs) produced by partner LuxI-type synthases. AHLs are most commonly QS autoinducers used by gram-negative bacteria. Exceptions to this are other QS signals, including autoinducer-2, diffusible signal factor, *Pseudomonas* quinolone signal and new molecules [2,3]. Gram-positive bacteria do not harbor LuxI/R homologues and instead utilize unmodified or modified oligopeptides as autoinducers. At present, many known autoinducers are bound by a membrane-bound sensor kinase located in the cell inner membrane, which switches its phosphatase and kinase activity in response to interaction with peptides, which changes the phosphorylation state of bacterial cognate response regulators and finally leads to activation or inhibition of QS target genes [4]. Both gram-positive and gram-negative bacteria use the QS system, and interfering with QS has been identified as a potential novel targeted therapeutic strategy to treat bacterial infections [5,6,7,8]. For example, gram-negative bacterial QS inhibition by QSIs is depicted in Figure 1. We display different mechanisms of action against a QS system; (a) inhibition of autoinducer synthases or decrease of activity of receptor proteins; (b) inhibition of autoinducer biosynthesis; (c) degradation of autoinducers; and (d) interference with binding of autoinducers and receptor proteins by competitive binding of autoinducer analogues and receptor proteins. For pathogens that regulate virulence via signaling molecules, QS interference also renders bacterial infections more benign and promotes the host innate immune system to more effectively eradicate the pathogens.

Marine microbial species, due to marine chemodiversity, have been considered as an untapped source for unique chemical leads with numerous biological activities [9,10,11]. These compounds provided a wide range of valuable drug candidates for treating various diseases in plants, animals and humans. However, marine microbial species have not been fully exploited like their terrestrial counterparts; according to the statistics, valuable compounds derived from marine environments have been discovered to a much lower extent (1%) than terrestrial environments so far, suggesting the very low percentage of metabolites isolated from marine microbial species [12]. Also, some evidence suggests that QS is a frequent phenomenon in marine environments [13]; QSIs have been found in diverse marine microbial species, such as marine bacteria, actinomycetes and fungi. The aim of this review is to give a comprehensive overview of QSIs from marine microbial species and their synthetic derivatives with QS inhibitory activity.

## 2. QSIs from Marine Bacteria and Their Derivatives with QS Inhibitory Activity

### 2.1. QSIs from Marine Gram-Positive Bacteria and Their Derivatives with QS Inhibitory Activity

Halophilic microorganisms possess a multitude of bioactive secondary metabolites due to their unique physiological and genetic properties. *Halobacillus salinus* C42 from a sea grass sample collected in the Point Judith Salt Pond, South Kingstown, RI afforded two phenethylamide metabolites, 2,3-methyl-*N*-(2’-phenylethyl)-butyramide (**1**) and *N*-(2’-phenylethyl)-isobutyramide (**2**), which were proven nontoxic to a panel of bacteria, fungi and microalgae [14,15]. These compounds inhibited QS-regulated violacein biosynthesis of *Chromobacterium violaceum* CV026 and green fluorescent protein production of *Escherichia coli* JB525. They acted as antagonists of bacterial QS by competing with AHL for receptor binding. The *cyclo*(L-Pro-L-Val) (**3**) isolated by *Haloterrigena hispanica* SK-3 could promote the expression of QS-regulated genes in bacterial AHL reporters, suggesting that archaea have the ability to interact with AHL-producing bacteria in syntrophic communities [16]. In contrast, four different diketopiperazines (DKPs): *cyclo*(L-Pro-L-Phe) (**4**), *cyclo*(L-Pro-L-Leu) (**5**), *cyclo*(L-Pro-L-*iso*Leu) (**6**), and *cyclo*(L-Pro-D-Phe) (**7**) isolated from *Marinobacter* sp. SK-3 demonstrated their QS-inhibitory activities based on the test of *C. violaceum* CV017 and *E. coli* [17]. This indicated that DKPs from microorganisms could activate or inhibit bacterial QS, pointing to a vital role of these molecules within microbial communities.

Three active metabolites isolated from *Oceanobacillus* sp. XC22919 were identified as 2-methyl-*N*-(2′-phenylethyl)-butyramide (**8**), 3-methyl-*N*-(2′-phenylethyl)-butyramide (**9**) and benzyl benzoate (**10**), and were first reported to exhibit the apparent QS inhibitory activities against *C. violaceum* 026 and *Pseudomonas aeruginosa* [18]. These molecules could inhibit violacein production in *C. violaceum* 026, as well as pyocyanin production, elastase and proteolytic enzymes, and biofilm formation in *P. aeruginosa*. Among them, Compound **8** significantly inhibited the formation of biofilm of *P. aeruginosa*, with a maximum of 50.6% inhibition, at 100 μg/mL. Saurav et al. [19] performed bioassay-guided isolation from three bacterial isolates of sponges (*Nautella* sp., *Erythrobacter* sp. CUA-870, and *Dietzia maris* IHBB 9296). The isolates Cc27, Pv86 and Pv91were found to be positive for QS inhibitory activity and inhibited the formation of biofilm by over 50% in tested strains (*E. coli*, *P. aeruginosa* PAO1, and *Bacillus subtilis*). Finally, nine main secondary metabolites (**11**–**19**) were identified in Cc27 (**11**–**13**), Pv86 (**14**–**16**), and Pv91 (**17**–**19**) using LC–HRMS/MS.

Two novel depsipeptides, solonamide A (**20**) and B (**21**), from a marine *Photobacterium* were identified by bioassay-guided isolation [20]. They interfere with *agr* QS activity in the highly virulent, community-acquired strain USA300 and *Staphylococcus aureus* 8325-4. This is the first report of the *agr* QS inhibitors from the marine bacteria. Generally, the *S. aureus agr* QS system includes at least four *agr* subclasses, and the autoinducing peptide from each class could induce *agr* in strains of its own class rather than repress *agr* of other subclasses [21,22,23]. However, solonamide B reduced *agr* QS expression significantly in three of four known *agr* classes (*agr* group **I**, *agr* group **II**, *agr* group **IV**), as well as having a minor effect against *agr* group **III** in the *S. aureus agr* system. Moreover, solonamide B significantly decreased the expression of phenol-soluble modulins, directly controlled AgrA and the transcription of *agrA*, as well as dramatically reduced the overall toxicity of supernatants towards human neutrophils. This indicated that solonamide B not only interfered with the expression of AgrA and *agrA*, but also repressed biosynthesis of virulence factors controlled by the *agr* QS system [24]. Further analysis demonstrated that solonamide B interfered with *agr* QS activation by preventing interactions between AgrC sensor histidine kinase and *S. aureus* autoinducing peptides. Structural comparison of solonamide B and autoinducing peptides suggests that the ability to interfere with different *agr* QS classes is related to the cyclic structure of solonamide B, and the differences observed may correlate with the temporal RNAIII induction pattern or the individual structures of autoinducing peptides [25].

In order to further elaborate structure–function relationships for AgrC QS antagonists, an array of 27 lactam hybrid analogues based on solonamide B and autoinducing peptides were designed and tested for AgrC-inhibitory activity [26]. Among them, 21 compounds (**22**–**41**) showed inhibition on the *S. aureus* QS system. However, there was considerable difference for their inhibitory activity. Hybrid analogues with all-L stereochemistry of the amino acids (**22**) were equipotent to AgrC inhibitors, solonamides A and B; however, compounds **25**, **27** and **33–35** were 20- to 40-fold higher in the inhibition of AgrC than the starting hit compound **22**. The structure–activity relationship indicates several structural features are very important determinants for AgrC inhibition (Figure 2); (a) ring size must be identical to the known autoinducing peptides; (b) the tail should be preferably selected from short fatty acid moieties; (c) a Phe residue shows more potent inhibition than other aliphatic or aromatic residues in residue no. 2; (d) residues no. 3 and 4 need to be further studied as there are no clear-cut conclusions; (e) the Leu in residue no. 5 is also crucial for activity.

Morever, Kajerulff et al. [27] reported four novel *agr* QS-interfering cyclodepsipeptides, ngercheumicin F–I (**42**–**45**) from a marine *Photobacterium halotolerans*. All four ngercheumicins enhanced expression of *spa* and decreased transcription of *hla* and *rnaIII* in the *S. aureus lacZ* reporter assays. Further studies showed that ngercheumicins reduce expression of *rnaIII* in the CA-MRSA strain USA300 by Northern blot analysis, suggesting that ngercheumicins interfere with *agr* QS activation. It can be speculated that these compounds could interfere with QS pathways that exist in the marine environment or even act as a class of novel alternative QS molecules.

### 2.2. QSIs from Marine Gram-Negative Bacteria

Marine gram-negative bacteria have been discovered to produce QS inhibitory compounds. For example, *Vibrio alginolyticus* G16 from seaweed *Gracilaria gracilis* could disrupt QS signaling pathways and reduce biofilm formation in *Serratia marcescens*. An active compound, phenol, 2,4-bis(1,1-dimethylethyl) (**46**) was obtained and identified [28]. It could inhibit the QS-mediated virulence factor biosynthesis in *S. marcescens* and lead to a significant reduction in biofilm (85%), lipase (84%), haemolysin (70%), protease (42%), and extracellular polysaccharide (85%) without affecting bacterial growth. Quantitative PCR analysis confirmed that the *N*-butanoyl-L-homoserine lactone (C_4_-HSL)-mediated *bsmA* gene was obviously downregulated in *S. marcescens*. Reduction in expression level of *bsmA* could be related to the ability of phenol, 2,4-bis(1,1-dimethylethyl) to affect QS-regulated biofilm formation. In addition to anti-QS-mediated biofilm inhibition, phenol, 2,4-bis(1,1-dimethylethyl) simultaneously induced hydration of the microbial cell wall, which made it a potential anti-biofilm agent by dual approaches. At the same time, the compound also increased the susceptibility of *S. marcescens* to gentamicin, which opened another avenue for combination therapy to improve the effectiveness of clinical antibiotics.

Through bioassay-guided fractionation, Sun and coworkers [29] have recently obtained an active diketopiperazine, *cyclo*(Trp–Ser) (**47**) from the marine bacterium, *Rheinheimera aquimaris* QSI02. It decreased QS-regulated violacein biosynthesis (67%) of *C. violaceum* CV026 and pyocyanin biosynthesis (65%), elastase activity (40%) and biofilm formation (60%) of *P. aeruginosa* PAO1. The analysis of molecular dynamics suggested that *cyclo*(Trp–Ser) binds more easily to the LasR receptor than natural QS-signaling molecules (AHLs), but the opposite is true in the LasR receptor. These results demonstrated that *cyclo*(Trp–Ser) not only efficiently inhibited the biosynthesis of violacein in *C. violaceum* CV026, but also reduced the formation of biofilm and other QS-mediated phenotypes in *P. aeruginosa* PAO1. In addition to these small molecules, MomL (**48**), a novel AHL lactonase derived from marine *Muricauda olearia* Th120, also significantly attenuated the virulence factor production (protease and pyocyanin) of *P. aeruginosa* PAO1. The growth of PAO1 was almost not affected, whereas AHL accumulations in the cultures were obviously reduced, suggesting that the reduction of production of virulence factors was due to AHL degradation [30]. Although QS processes were widely distributed in marine microorganisms and QSIs were discovered as a frequent phenomenon in the marine environments [31], QSIs from marine gram-negative bacteria are still very scarce when compared to marine gram-positive bacteria, and there may be drawbacks to probing QSIs in marine gram-negative bacteria using traditional bioassay-guided isolation.

## 3. QSIs from Marine Actinomycetes and Their Derivatives with QS Inhibitory Activity

As recently reported by Fu and coworkers [32], three new α-pyrones, nocapyrones H (**49**), I (**50**) and M (**51**) from marine-derived actinomycete *Nocardiopsis dassonvillei* subsp. *dassonvillei* XG-8-1 exhibited inhibitory activities on QS-controlled gene expression in both *C. violaceum* CV026 and *P. aeruginosa* QSIS-*lasI* biosensors at a concentration of 100 μg/mL. This is the first report of α-pyrones inhibiting QS-regulated gene expression in pathogenic bacteria. Recently, four novel α-pyrones and eight known analogues were also detected in the secondary metabolites of *Streptomyces* sp. OUCMDZ-3436 isolated from the marine green alga *Enteromorpha prolifera* [33]. The results of bioassays suggested that these α-pyrones did not exhibit any QS inhibitory activity. However, the skeleton of α-pyrone could be easily transformed into pyridine-2(1*H*)-one, which had been proved to have a variety of biological activities [34,35,36,37,38,39,40,41,42]. Therefore, based on a diversity-enhanced extracts approach [43,44], four novel α-pyridones (**52–55**) were obtained and exhibited the inhibitory effect on gene expression regulated by QS in *P. aeruginosa* QSIS-*lasI* biosensors at 6.35 μg/well.

In order to further obtain potent QSIs from pyrone-derived compounds, Park and coworkers [45] designed and synthesized several novel pyrone-derived QSIs (**56–65**) to inhibit the binding of *N*-(3-oxododecanoyl)-L-homoserine lactone (OdDHL) to the LasR of *P. aeruginosa*. Among the 10 novel pyrone-derived QSIs, compound **63** exhibited the most potent in-vitro inhibitory activity against biofilm formation. Furthermore, all of the selected QS-inducible genes, including synthase genes (*lasI*, *rhlI*, *pqsC*, *pqsD*, *pqsH*, and *phnB*), and auto-inducer receptor genes (*lasR*, *mvfR*, and *rhlR*), were also significantly downregulated by compound **63**. The modeling studies indicated that it mostly interacted with residues in the binding pocket of LasR that was highly similar to the crystal ligand OdDHL. The structure–activity relationship indicated that the pyrone derivatives with more than nine alkyl chains would significantly reduce biofilm formation.

*Cyclo*(Pro–Gly) (**66**) and cinnamic acid (**67**) from marine invertebrate-derived *Streptomyces* sp. were reported to attenuate *P. aeruginosa* virulence as QS inhibitors [46]. The in-vivo study suggested that cinnamic acid protected *Caenorhabditis elegans* from the virulence of *P. aeruginosa* resulting in reduced mortality [47]. This protective mechanism is likely to be a consequence of competitive suppression of RhlR and LasR receptor proteins by cinnamic acid. A significant reduction in colonization of the bacteria treated with cinnamic acid was also further observed in the nematode. These data were consistent with past results that curcumin increased the survival rate of *Caenorhabditis elegans* ~28% by decreasing the expression of genes involved in biofilm formation and attenuating the biosynthesis of signal molecules in *P. aeruginosa* PAO1 [48]. The in-silico analysis showed that it could act as a competitive inhibitor for the natural signal molecules towards active pockets of LasR and RhlR QS circuits in *P. aeruginosa.*

Several derivatives structurally related to cinnamic acid were also synthesized and assessed for their effects on the QS process. Cinnamyl alcohol (**68**), methyl *trans*-cinnamate (**69**) and allyl cinnamate (**70**) were able to completely inhibit *C. violaceum* QS at 1 mM, and cinnamamide (**71**), 4-chlorocinnamic acid (**72**), α-methylcinnamic acid (**73**) and 3,4-(methylenedioxy)cinnamic acid (**74**) at 5 mM [49]. These data were consistent with past results that **71** and **72** inhibited *Vibrio* spp. virulence factor biosynthesis in vitro and in vivo [50,51]. However, by contrast, **68** and **69** were devoid of inhibitory activity as QSIs against *Vibrio* spp. [50]. This suggests that they are selective QSIs against different pathogens. The structure–activity relationship of **67** indicated that the replacement of the carboxylic acid moiety by an aldehyde group (cinnamaldehyde) or the lack of the double bond (3-phenylpropionic acid) leads to inactivity. The replacement of the vinyl acid function by a vinyl sulfone resulted in a more active compound, methyl-styryl sulfone (**75**). Substituents with electron withdrawing properties increased QS activity. The inhibitory activity decreased in the order 4-trifluoromethyl cinnamic acid (**76**) > **72** > 2,3,4,5,6-pentafluoro-cinnamic acid (**77**). The aromatic ring of cinnamic acid was replaced by an alkyl group, and a carboxylic acid moiety was replaced by an acrolein moiety, and these changes still led to active autoinducer-2 signal QS inhibitors, such as *(E)*-2-pentenal (**78**), *(E)*-2-tridecenal (**79**), methyl-*(E)*-2-nonenoate (**80**), and *(E)*-2-heptenal (**81**) [50].

The research group of Miao et al. [52] obtained and identified a secondary metabolite, actinomycin D (**82**) derived *Streptomyces parvulus* HY026 isolated from a seawater sample, which showed remarkable anti-QS activity. It significantly inhibited the violacein biosynthesis of *C. violaceum* (65%) at 12.5 μg/mL and prodigiosin production (the pigment inhibition zone of 13.5 mm) of *Serratia proteamaculans* 657 at 25 μg/disc without affecting bacterial growth. These findings indicated that researchers not only pay attention to the discovery of novel compounds, but also point to known antibiotics for the discovery of new valuable bioactivity.

## 4. QSIs from Marine Fungi and Their Derivatives with QS Inhibitory Activity

Marine fungi were targeted as potent producers of QSIs, since they possess the ability to synthesize and secrete diverse secondary metabolites, such as peptides, terpenes, polyketide-derived alkaloids, and mixed biosynthesis metabolites [53]. A γ-pyrone derivative, kojic acid (**83**) from marine-derived fungus *Altenaria* sp. isolated from marine green alga *Ulva pertusa* of Pyoseon Beach, Jeju Island, inhibited QS-dependent luminescence of the reporter *E. coli* pSB401 induced by *N*-hexanoyl-L-homoserine lactone (C_6_-HSL) at >36 μM. However, the molecule only interfered with LuxR reporters [54]. Equisetin (**84**), from a marine fungus *Fusarium* sp. Z10, inhibited biofilm formation and swarming motility of *P. aeruginosa*. Further studies showed that the compound inhibited elastase of *P. aeruginosa* PAO1 and transcriptional activation of *lasB* in *E. coli* MG4/pKDT17, attenuated pyocyanin biosynthesis of *P. aeruginosa* PAO1 and transcriptional activation of *PqsA* in *E. coli* pEAL08-2, and declined rhamnolipid biosynthesis, swarming motility and transcriptional activation of *rhlA* in *E. coli* pDSY. These data indicated that equisetin could inhibit *las*, *rhl* and the *PQS* system [55]. Asteltoxin (**85**), a known QS inhibitor from a marine fungus *Penicillium* sp. QF046, exhibited more potent inhibition of violacein than positive control, (*Z*-)-4-bromo-5-(bromomethylene)-2(*5H*)-furanone, and decreased the expression of multiple QS-related genes (*lasA*, *lasB*, *vioB*, *vioI*, *cynS*, and *hcnB*) [56]. A new bacterial enoyl-acyl carrier protein reductase inhibitor, meleagrin (**86**), from the marine fungus *Penicillum chrysogenium* isolated from seashore slime of Daechun beach, Chungcheongnam-do, Korea, inhibited QS of *C. violaceum* CV017 with minimum inhibitory concentration (MIC) of 138.42 mM [57,58]. The fermentation broth of *Penicillium* sp. SCS-KFD08 isolated from the marine animal *Sipunculus nudus* collected from Haikou Bay, China led to the isolation of six QSIs (**87–92**), based on the test of biosensor *C. violaceum* CV026. These secondary metabolites exhibited obvious anti-QS activity against *C. violaceum* CV026 at a dosage of 50 μg/well. Among them, compounds **91** and **92** inhibited violacein biosynthesis in *C. violaceum* CV026 cultures induced by signal molecule C_6_-HSL by up to 46% and 49% at subminimal inhibitory concentration (sub-MIC) of 300 μM, respectively [59].

In the screening of QS–disrupting molecules, 75 marine fungal isolates were obtained from saline lakes, mangrove rhizosphere and reef organisms. Their QS inhibitory activity was evaluated using *C. violaceum* CV026. Four strains of endophytic fungi belonging to *Sarocladium* (LAEE06), *Fusarium* (LAEE13), *Epicoccum* (LAEE14), and *Khuskia* (LAEE21) exhibited potent activity at concentrations ranging from 50 to 500 μg/mL. LC–HRMS analysis of these fungal bioactive metabolites resulted in the identification of several major compounds whose QS inhibitory properties had been known or unknown so far (**93**–**108**) [60]. For example, fusaric acid (**96**) and linoleic acid (**108**) were isolated from the samples of *Fusarium* (LAEE13) and *Khuskia* (LAEE21), respectively, and their ability to interfere with the QS system had been previously reported [61,62,63,64]. Two major components isolated from *Epicoccum* (LAEE14) were tentatively classified as two DKPs, variecolorin N (**104**) and phenylahistin (**105**). They may act as QS agonists or antagonists due to the wide recognition of DKPs as QSIs [65].

Facile and expeditious synthetic strategies have proven to be an efficient tool in obtaining valuable QSIs. Taking advantage of microwave–assisted synthesis, 39 fusaric acid analogues were obtained and tested for their QS inhibitory activity in three QS screening models, *luxI-gft*, *lasB-gft*, and *rhlA-gft* [66]. In the *luxI-gft* QS system, compounds **109**–**114** revealed QS inhibition at concentrations from 6.25 to 100 μg/mL. Compound **115** exhibited a little QS inhibition at 3.13 μg/mL whereas at higher concentrations (6.25–100 μg/mL) its QS inhibition was obviously enhanced. In the *lasB-gft* QS system, compound **116** showed good QS inhibition from 125 μg/mL. A structure–activity relationship for QS inhibition is depicted in Figure 3; (a) the C-2 ester group is essential for inhibition of QS, since it can simulate the intermolecular interaction solicited by the lactone moiety of QS signal molecules; (b) the carboxylic acid substituent at C-2 shows no QS inhibitory activity and inhibits the growth of bacteria; and (c) the alkyl substituent at C-5 is not mandatory, it can be swapped by an aromatic/heterocyclic aromatic ring or alkoxy, even if the compounds exhibit modest QS inhibitory activity.

## 5. Conclusions and Perspectives

Through reviewing the literature of QSIs from marine microorganisms and their synthetic analogues, it is very clear that 116 QSIs are derived from marine microorganisms and their synthetic analogues (Table 1). Therefore, more effort needs to be made towards the promising strains from marine microorganisms for the discovery of novel QSIs. Moreover, the majority of QSIs from marine microorganisms and their synthetic analogues identified to date function as useful chemical probes for mechanistic or structural studies rather than lead-like compounds for further anti-infective drug development.

In addition, the current lack of methodological standardization in assessing effectiveness of QSI candidates limits the broad validity of any findings. The use of only laboratory-adapted strains may lead to a major pitfall in this field, since these strains may be distant from pathogens of relevant clinical infections. Therefore, further screening of QSIs in conditions that mimic in-vivo pathogenic infections will be vital for the future study and development of QSIs. It is worth noting that QSIs are most likely to be beneficial as potent adjuvants of conventional antibiotics for the treatment of clinical treatment rather than as standalone therapeutic agents, as QSIs allow for temporal control of virulence gene expression rather than result in selective pressures on bacterial survival [67,68,69,70]. For example, the QSI pyrizine–2–carboxylic acid significantly increased the susceptibility of antibiotics (tetracycline, doxycycline, erythromycin and chloramphenicol) against multidrug-resistant *Vibrio cholerae* [71]. Undoubtedly, simultaneous use of inhibitors for various targets including the QS system will contribute to combating multiantibiotic–resistant bacteria [72,73].

## Figures and Tables

**Figure 1 marinedrugs-17-00080-f001:**
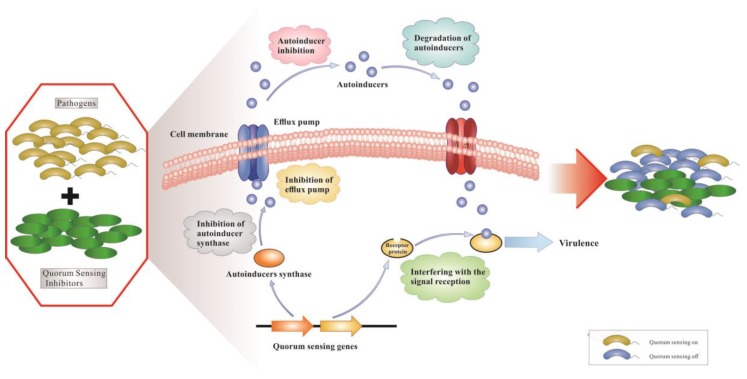
The mechanisms of action of QSIs in gram-negative pathogens.

**Figure 2 marinedrugs-17-00080-f002:**
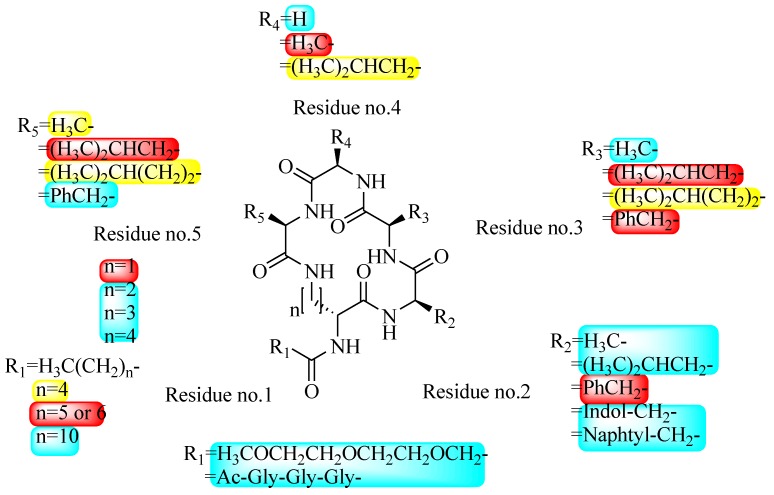
Structure–activity relationships obtained lactam hybrid analogues with AgrC-inhibitory activity. Color coding stands for the effect of each of the structure features: red (high potency); yellow (low potency); blue (detrimental to potency).

**Figure 3 marinedrugs-17-00080-f003:**
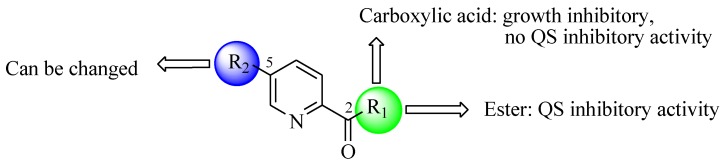
Structure–activity relationships obtained for fusaric acid analogues.

**Table 1 marinedrugs-17-00080-t001:** QSIs from marine microorganisms.

No.	Structures of QSIs	Source	Biosensor Microorganisms	Specific Inhibitory Activity	Ref.
**1**	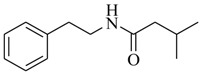	*H. salinus* C42	*V. harveyi* BB120	Inhibition of violacein of *C. violaceum* CV026 and green fluorescent protein of *E. coli* JB525	[14,15]
**2**	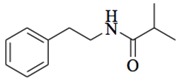
**3**	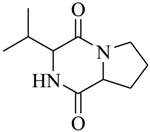	*H. hispanica* SK-3	*E. coli A. tumefaciens* NTL4	Inhibition of green fluorescent protein of *V. anguillarum* DM27	[16]
**4**	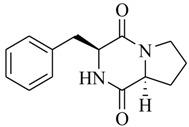	*Marinobacter* sp. SK-3	*C. violaceum* CV017	Inhibition of luminescence of *E. coli* pSB401	[17]
**5**	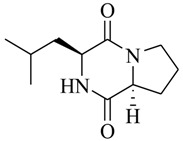
**6**	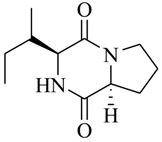
**7**	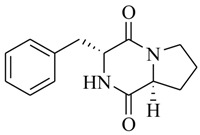
**8**	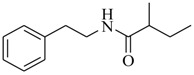	*Oceanobacillus* sp.XC22919	*C. violaceum* CV026	Inhibition of violacein of *C. violaceum* ATCC12472 and pyocyanin, elastase and proteolytic activities, and biofilm formation of *P. aeruginosa*	[18]
**9**	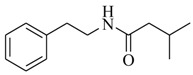
**10**	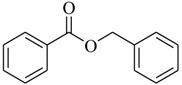
**11**	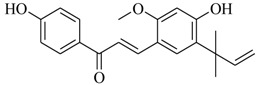	*Nautella* sp.	*C. violaceum* CV026	Inhibition of pyoyanin production of *P. aeruginosa*PAO1, and biofilm formation in *E. coli*, *P. aeruginosa* PAO1, and *Bacillus subtilis*	[19]
**12**	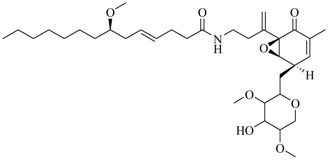
**13**	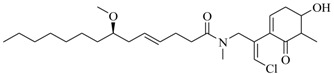
**14**	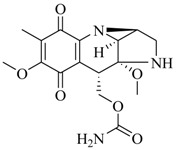	*Erythrobacter* sp. CUA-870	*C. violaceum* CV026	Inhibition of pyoyanin production of *P. aeruginosa* PAO1, and biofilm formation in *E. coli*, *P. aeruginosa* PAO1, and *Bacillus subtilis*	[19]
**15**	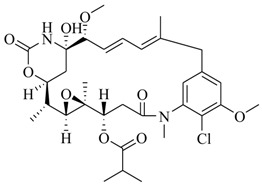
**16**	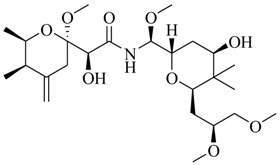
**17**	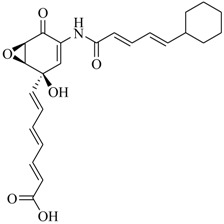	*Dietzia maris* IHBB 9296	*C. violaceum* CV026	Inhibition of pyoyanin production of *P. aeruginosa* PAO1, and biofilm formation in *E. coli*, *P. aeruginosa* PAO1, and *Bacillus subtilis*	[19]
**18**	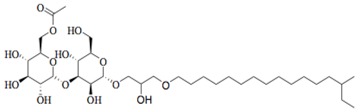
**19**	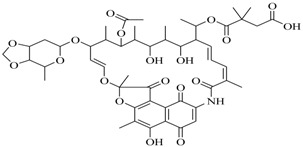
**20**	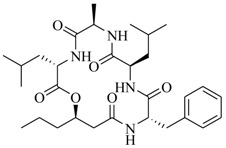	*Photobacterium*	*S. aureus* 8325-4	Inhibition of *agrA* and AgrA, and virulence factors α-hemolysin and Protein A in *S. aureus*	[20]
**21**	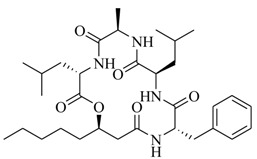
**22**: n = 1; R=H_3_C(CH_2_)_5_-**23**: n = 1; R=H_3_C(CH_2_)_8_-**24**: n = 2; R=H_3_C(CH_2_)_5_-	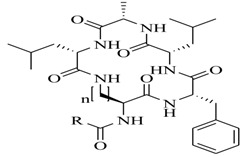	Chemical synthesis based on solonamide B and autoinducing peptides	P3-*blaZ* reporter strain	Inhibition of β-actamase activity controlled by *S. aureus agr* QS system	[26]
**25**: R=H_3_C(CH_2_)_5_-; R_1_=H_3_C-**26**: R=H_3_C(CH_2_)_4_-; R_1_=H_3_C-**27**: R=H_3_C(CH_2_)_6_-; R_1_=H_3_C-**28**: R=H_3_C(CH_2_)_10_-; R_1_=H_3_C-**29**: R=H_3_C(CH_2_)_5_-; R_1_=H**30**: R=H_3_C(CH_2_)_5_-; R_1_=(H_3_C)_2_CHCH_2_-	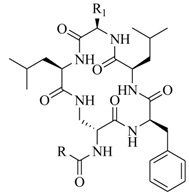
**31**: R=H_3_C-; R_1_=(H_3_C)_2_CHCH_2_-**32**: R=(H_3_C)_2_CHCH_2_-; R_1_=H_3_C-**33**: R=PhCH_2_-; R_1_=(H_3_C)_2_CHCH_2_-**34**: R=(H_3_C)_2_CHCH_2_CH_2_-R_1_=(H_3_C)_2_CHCH_2_-**35**: R=(H_3_C)_2_CHCH_2_-R_1_=(H_3_C)_2_CHCH_2_CH_2_-	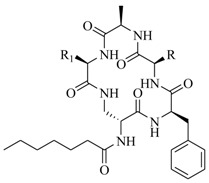
**36**	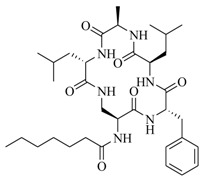
**37**	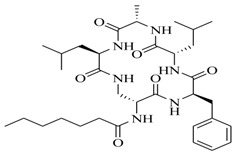
**38**	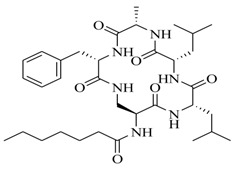
**39**	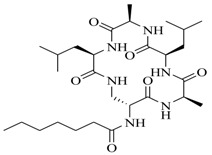
**40**: R= 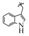 **41**: R= 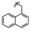	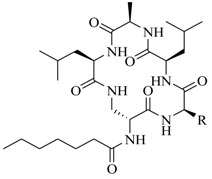
**42**: R=CH_3_(CH_2_)_5_CH=CH(CH_2_)_3_-**43**: R=CH_3_(CH_2_)_10_-**44**: R=CH_3_(CH_2_)_5_CH=CH(CH_2_)_5_-**45**: R=CH_3_(CH_2_)_12_-	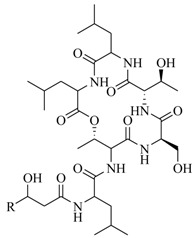	*P. halotolerans*	*S. aureus lacZ* reporter	Inhibition of expression of *rnaIII* in community-associated methicillin-resistant *S. aureus* USA300	[27]
**46**	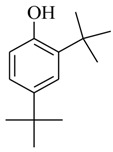	*V. alginolyticus* G16	*S. marcescens*	Inhibition of biofilm, protease, haemolysin, lipase, prodigiosin and extracellular polysaccharide secretion in *S. marcescens*	[28]
**47**	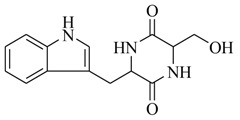	*R. aquimaris* QSI02	*C. violaceum* CV026	Inhibition of pyocyanin production, elastase activity and biofilm formation in *P. aeruginosa* PA01	[29]
**48**	*N*-acylhomoserine lactonase	*Tenacibaculum* sp. 20J	*C. violaceum* CV026	Inhibition of *SdiA* in *E. coli*	[30]
**49**	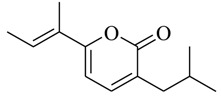	*Nocardiopsis dassonvillei* subsp. *dassonvillei* XG-8-1	*C. violaceum* CV026 and *P. aeruginosa* QSIS-*lasI* biosensors	Inhibition of *C. violaceum* CV026 and *P. aeruginosa* QSIS-*lasI* biosensors	[32]
**50**	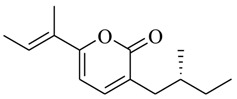
**51**	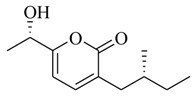
**52**	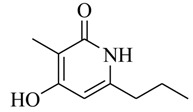	*Streptomyces* sp. OUCMDZ-3436	*P. aeruginosa* QSIS-*lasI* biosensors	Inhibition of gene expression controlled by QS in *P. aeruginosa* QSIS-*lasI* biosensors	[33]
**53**	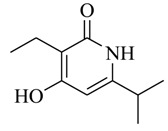
**54**	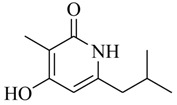
**55**	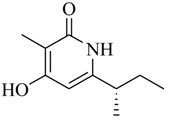
**56**: n = 3; **57**: n = 4; **58**: n = 5; **59**: n = 6; **60**: n = 7; **61**: n = 8; **62**: n = 9; **63**: n = 10; **64**: n = 11; **65**: n = 12	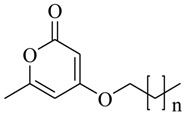	Chemical synthesis based on pyrones	*P. aeruginosa*	Inhibition of biofilm formation of *P. aeruginosa*	[45]
**66**	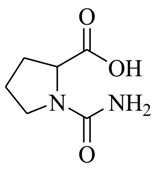	*Streptomyces* sp.	*C. violaceum* ATCC 12472and *P. aeruginosa* ATCC 27853	Inhibition of swarming, pyocyanin, biofilm formation, rhamnolipid production in *P. aeruginosa*	[46]
**67**	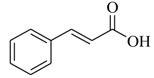
**68**	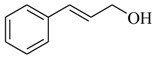	Chemical synthesis based on cinnamic acid	*C. violaceum*	Inhibition of violacein of *C. violaceum*	[49]
**69**	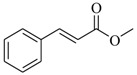
**70**	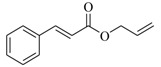
**71**	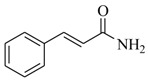
**72**	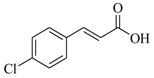
**73**	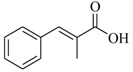
**74**	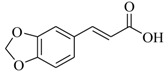
**75**	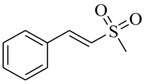	Chemical synthesis based on cinnamic acid	*Vibrio* spp	Inhibition of biofilm formation, pigment production and protease production in *Vibrio* spp.	[50]
**76**	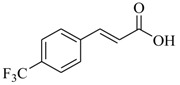
**77**	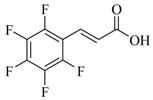
**78**: R_1_=H; R_2_=C_2_H_5_-**79**: R_1_=H; R_2_=C_10_H_21_-**80**: R_1_=CH_3_O-; R_2_=C_6_H_13_-**81**: R_1_=H; R_2_=C_4_H_9_-	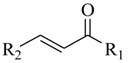
**82**	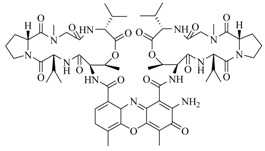	*S. parvulus* HY026	*C. violaceum* ATCC 12472	Inhibition of violaceim of *C. violaceum* and prodigiosin production of *Serratia proteamaculans* 657	[52]
**83**	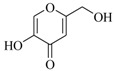	*Altenaria* sp.	*E. coli* pSB401	Inhibition of luminescence of *E. coli* pSB401	[54]
**84**	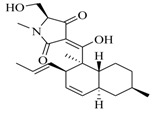	*Fusarium* sp. Z10	*P. aeruginosa* QSIS- *lasI* biosensor	Inhibition of formation of biofilm, swarming motility, and the production of virulence factors in *P. aeruginosa.*	[55]
**85**	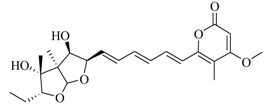	*Penicillium* sp. QF046	*C. violaceum* CV026	Inhibition of violacein of *C. violaceum* CV026	[56]
**86**	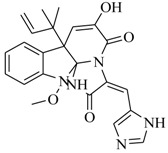	*P. chrysogenium*	*C. violaceum* CV017	Inhibition of *S. aureus* FabI	[57,58]
**87**	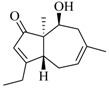	*Penicillium* sp. SCS-KFD08	*C. violaceum* CV026	Inhibition of violacein of *C. violaceum* CV026	[59]
**88**	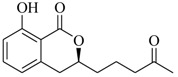
**89**	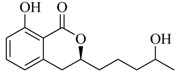
**90**	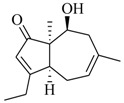
**91**	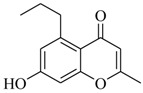
**92**	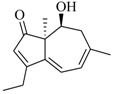
**93**	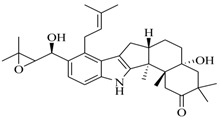	*Sarocladium*	*C. violaceum* CV026	Inhibition of violacein of *C. violaceum* CV026	[60]
**94**	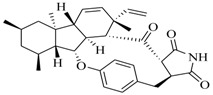
**95**	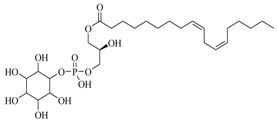
**96**	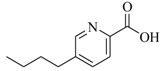	*Fusarium*	*C. violaceum* CV026	Inhibition of violacein of *C. violaceum* CV026	[60]
**97**	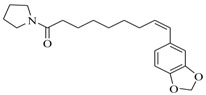
**98**	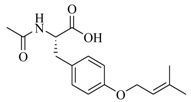
**99**: R=CH_3_-**100**: R=H	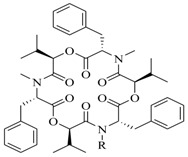
**101**	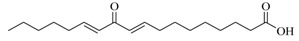
**102**	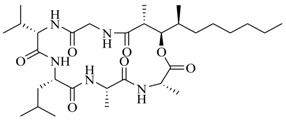	*Epicoccum*	*C. violaceum* CV026	Inhibition of violacein of *C. violaceum* CV026	[60]
**103**	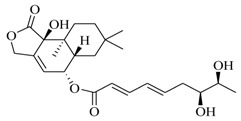
**104**	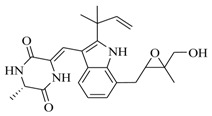
**105**	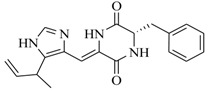
**106**	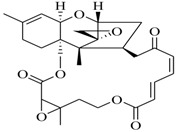	*Khuskia*	*C. violaceum* CV026	Inhibition of violacein of *C. violaceum* CV026	[60]
**107**	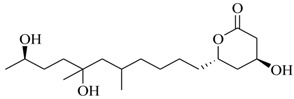
**108**	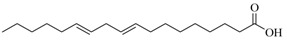
**109**: R_1_= 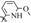 ; R_2_=CH_3_O-**110**: R_1_= 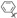 ; R_2_=CH_3_CH_2_O-**111**: R_1_=CH_3_CH_2_CH_2_-; R_2_=CH_3_O-**112**: R_1_= 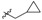 ; R_2_=CH_3_O-**113**: R_1_= 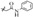 ; R_2_=CH_3_O-**114**: R_1_=CH_3_CH_2_CH_2_CH_2_-R_2_=CH_3_CH_2_O-**115**: R_1_=CH_3_CH_2_CH_2_CH_2_-; R_2_=HO-**116**: R_1_=Br-; R_2_=CH_3_O-	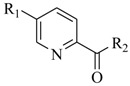	Chemical synthesis based on fusaric acid	*P. aeruginosa* and *Vibrio fischeri*	Inhibition of the *las* and *rhl* QS system in *P. aeruginosa* and the *lux* QS system in *Vibrio fischeri*	[66]

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
