# Peer review of "Quorum Sensing Inhibitors from Marine Microorganisms and Their Synthetic Derivatives"

_marinedrugs, 2019, doi:10.3390/md17020080_

Round 1
Reviewer 1 Report
The authors should improve the manuscript. All suggestion and corrections are reported in the attached file.

Author Response
Dear editor and reviewer 1
Please see attachment about response to reviewer 1 comments.

Reviewer 2 Report
In the present review, the authors have summarized the different quorum sensing inhibitory molecules identified from marine microorganisms. They have performed an exhaustive analysis of the bibliography linked to this subject up to day, to give a global image of the state of the art of this interesting and promising research field. However, in the way that the review is written, it is an enumeration of the different molecules that have been identified without any further discussion or interpretation of their importance. This is crucial in order to considerably increase the interest of the work. More, several major issues should be addressed:
1- In the Introduction, the authors briefly present the quorum sensing phenomenon; however, the description that they use is mainly focused in the gram-negative model. In fact, in the references that they cited they only included those focalized in gram negative bacteria. For example, in lines 27-28, they describe the process as “These molecules are released into the surrounding environment where they could be recognized by cognate receptors located at the bacterial cell wall”. This is not representative of all cases due to in some gram-positive QS systems such as RRNPP family in Streptococcaceae, the signal molecule in form of peptide is actively re-imported inside the bacterium.
2- Quorum quenching process (the disruption of QS signaling) encompasses very diverse phenomena and mechanisms which has not been presented in this review. Even if the authors are focalized in QS inhibitor molecules, they should unless show other possible strategies and justify their choice to focus their attention in QSI molecules.
3- The Figure 1 of the manuscript is really very interesting and well built. Though, the authors didn’t explain and discuss the different strategies in which the QSI could act based on this Figure.
4- A table summarizing the QSI including the bacterial species from which have been isolated, the specific inhibitory activity as well as the method utilized for their activity characterization (i.e. biosensor microorganism) could be very useful to better understand their relevance and possible applications. Further, this could help also to lighten up the listing of all the examples in the text, and better explain the molecular or chemical mechanisms involved in each case.
5- In the section 2.1. (“QSIs from marine Gram-positive Bacteria and their Derivatives with QS inhibitory activity”), the archaeon Haloarcula hispanica has been included. This species should be removed from this section or change its name.
6- It is not very useful to include in the main text the chemical structure of different QSI compounds (Figures 2, 4, 5 and 6). They could be included as supplementary material.
7- The first time that is mentioned the name of a species, the complete name should be written. All along the manuscript there are several mistakes concerning the species names (i.e. V. cholerae in line 40, Serratia marcescens in line 145, etc.). These mistakes might be corrected in all the text.
Minor comments:
- Lines 81-82: authors stated that “9 main secondary metabolites were identified in…”; why these molecules are interesting or important? They should further discuss this data.
- Line 101: “an array” instead of “An array”.
- Line 116: “assays. Further” instead of “assays, further”.
- Line 144: authors stated “Further studies…”; however, there is no any reference supporting this data.
- Line 147: “simultaneously” instead of “stimultaneostly”.
- Line 178: “Streptomyces” instead of “Streptomycs”.
- Line 182: this sentence is complex to understand; I suggest: Therefore, based on “diversity-enhanced extracts” approach…
- Line 188: “activity” instead of “activities”.
- Line 298: the word “only” should be removed due to 117 QSI molecules is an important number of chemical compounds.
- Line 307: please, remove one “that” because it is duplicated.
- Line 313: “multiantibiotic bacteria” means nothing. It should be read “multiantibiotic resistant bacteria”.
Author Response
Dear editor and reviewer 2
Please see attachment about response to reviewer 2 comments

Round 2
Reviewer 2 Report
In the current revised version of the review entitle “Quorum Sensing Inhibitors from Marine Microorganisms and Their Synthetic Derivatives”, the authors have performed an exhaustive improvement of their manuscript, considering all the comments and suggestions that I have made. My only comment is that I did not found the Table 1 that they mention in their answer to my comments in the manuscript and there is not any reference to this table, so it is not possible to take it in consideration. I would like to have the possibility to review this missed information.
Author Response
Dear editor,
I increased Table 1 for response to reviewer 2 comments.
A table summarizing the QSI including the bacterial species from which have been isolated, the specific inhibitory activity as well as the method utilized for their activity characterization (i.e. biosensor microorganism) could be very useful to better understand their relevance and possible applications. Further, this could help also to lighten up the listing of all the examples in the text, and better explain the molecular or chemical mechanisms involved in each case.
Response: I have increased a table (Table 1) including structures, sources, biosensor microorganisms, and specific inhibitory activity.